# The Transformative Possibilities of the Microbiota and Mycobiota for Health, Disease, Aging, and Technological Innovation

**DOI:** 10.3390/biomedicines7020024

**Published:** 2019-03-28

**Authors:** Lucas Jones, Jessica Kumar, Adil Mistry, Thriveen Sankar Chittoor Mana, George Perry, V. Prakash Reddy, Mark Obrenovich

**Affiliations:** 1Geriatric Research, Education, and Clinical Center, Cleveland Veterans Affairs Medical Center, Cleveland, OH 44106, USA; ldj20@case.edu (L.J.); jessicakumardo@gmail.com (J.K.); 2Department of Molecular and Microbiology, Case Western Reserve University School of Medicine, Cleveland, OH 44106, USA; 3Departments of Engineering and Chemistry, Cleveland State University, Cleveland, OH 44115, USA; adilamistry@gmail.com; 4Division of Infectious Diseases and HIV Medicine, Case Western Reserve University, Cleveland, OH 44106, USA; thriveen.sankar@gmail.com; 5Distinguished University Chair in Neurobiology, The University of Texas at San Antonio, San Antonio, TX 78249, USA; George.Perry@utsa.edu; 6Missouri University of Science and Technology, Rolla, MO 65409, USA; 7Department of Chemistry, Case Western Reserve University, Cleveland, OH 44106, USA; 8MD and CSO, the Gilgamesh Foundation.org, Cleveland, OH 44106, USA; 9Department of Physics, University of Toledo, Toledo, OH 43606, USA

**Keywords:** aging, microbiota, mycobiota, gut-brain-axis, CRISPR, blood–brain barrier, leaky gut, leaky brain, autism, schizophrenia, transsulfuration, synbiotics, parkinson disease, Alzheimer’s disease

## Abstract

The gut microbiota is extremely important for the health of the host across its lifespan. Recent studies have elucidated connections between the gut microbiota and neurological disease and disorders such as depression, anxiety, Alzheimer’s disease (AD), autism, and a host of other brain illnesses. Dysbiosis of the normal gut flora can have negative consequences for humans, especially throughout key periods during our lifespan as the gut microbes change with age in both phenotype and number of bacterial species. Neurologic diseases, mental disorders, and euthymic states are influenced by alterations in the metabolites produced by gut microbial milieu. We introduce a new concept, namely, the mycobiota and microbiota-gut-brain neuroendocrine axis and discuss co-metabolism with emphasis on means to influence or correct disruptions to normal gut flora throughout the lifespan from early development to old age. These changes involve inflammation and involve the permeability of barriers, such as the intestine blood barrier, the blood–brain barrier, and others. The mycobiota and microbiota–gut–brain axis offer new research horizons and represents a great potential target for new therapeutics, including approaches based around inflammatory disruptive process, genetically engineered drug delivery systems, diseased cell culling “kill switches”, phage-like therapies, medicinal chemistry, or microbial parabiosis to name a few.

## 1. Introduction

The human body exists in symbiosis with microbial communities on the skin, in the oral mucosa, vaginal mucosa, and in the gut [1]. The symbiotic relationship between the intestinal microbiota and gut immunity implies the necessity to keep constant gut surveillance controlling for excessive bacterial load and limiting pathogens. Changes in the abundance and type of organisms comprising the fecal microbiota extends beyond simple commensalism to influence health and disease through a wide variety of pathologies, which includes neurodegenerative disorders, digestive disease, hepatic diseases, diabetes, and viral infections [2,3,4,5,6]. 

Metagenomic studies have elucidated reductions in the richness of microbial genes and alterations to the functional capabilities of the fecal microbiota [7] leading to hallmarks of obesity, liver disease, and type II diabetes, which can be modified by dietary interventions [8,9]. The gut microbiome harbors over 150 times more genes than the human genome, which significantly increases the repertoire of functional genes possibly available to the host and contributes to system energy from food and energy harvesting from substrates that are indigestible by humans without bacterial digestive enzymes.

### 1.1. Relationship between the Brain and the Gut Microbiota

The gut microbiota is comprised of many diverse organisms each with their own niche in the gut. While each different grouping of bacteria has its own role in the gut, all the organisms in the gut flora produce metabolites and small molecules that can either have negative or positive impacts on host health [1]. The intestinal microbiome is colonized by organisms from the Firmicutes and Actinobacteria (Gram-positive), as well as Proteobacteria and Bacteroidetes (Gram-negative). However, in healthy individuals the majority of the gut is colonized by Firmicutes and Bacteroidetes. The gut flora produces small molecule metabolites including 40 known neurotransmitters, such as dopamine, serotonin, and GABA (gamma-aminobutyric acid, an inhibitory neurotransmitter). Typical roles of the gut flora include metabolizing nutrients such as plant phenols, sugars, and bile salts [2]. 

The metabolites and small molecule products produced by the gut flora are of extreme interest in the chemical signaling pathway called co-metabolism between the gut microbiota and the brain, named as the Microbiota–Gut–Brain (MGB) axis. Microbiota of the gut and the brain work in conjunction to promote proper function of the central nervous system. In healthy individuals, the relationship between the host and the gut microbiota is mutually beneficial, however, this relationship can be disrupted by the introduction of pathogenic bacteria, viruses, fungi, and other parasites. This ultimately leads to disruptions in the MGB axis and could have negative consequences on the mental health of the host. Change in the normal gut flora of the host can be attributed to a variety of different events: dietary changes, lifestyle changes, and antibiotic intervention [1]. Commensal bacteria in the gut work in conjunction with lymphoid tissues that are associated with the gut, to form a tight relationship between cognitive behavior and the gut microbiota [1,3]. Changes to the gut flora have an indirect impact on the behavior and brain physiology via metabolic signaling from the gut to the brain [1,3]. Intestinal microbiota have significant influence over the biochemistry of the host via production of hormones that can alter neurobiology [2]. This can influence the severity of diseases such as autism, Alzheimer’s disease, Parkinson’s disease, depression, anxiety, and a host of other mental disorders and diseases [2]. A healthy balanced gut microbiota is extremely important not only for the digestive system health, but also for mental and general health.

### 1.2. Blood Barriers and Their Roles in the MGB Axis

#### 1.2.1. Intestinal Blood Barrier

The intestinal blood barrier has two primary functions: first, is to act as a physical barrier to prevent the passage of harmful metabolites, microorganisms, and microbial toxins from getting into the blood stream; second, the intestinal blood barrier acts as a selectively permeable membrane to allow the transport of beneficial metabolites from the gut lumen to the blood stream [10]. Epithelial cells form a protective monolayer in order to mediate the barrier permeability. This epithelial monolayer is held together at the connecting edges of the epithelial cells by a host of connective proteins that form junctions. The three main junctions responsible for maintaining permeability are the desmosomes, tight junctions, and adherens junctions [10]. Tight junctions are on the apical side of the epithelial cell and are comprised of claudin binding and dimerization, which are mediated by adaptor proteins known as zona occludins. Another major protein player in this junction is the occludin proteins that also help mediate strong binding at this junction [10]. The major function of the tight junction is to seal the apical side gap between neighboring epithelial cells [10]. Adherens junctions are located more basal than the tight junctions, however, are not completely on the basal side of the membrane [10]. Adherens junctions are a cellular anchoring system that is used to connect the actin filaments of one cell to its neighboring cells [11]. Adherens junctions utilize several adaptor proteins to mediate the hemophilic binding of classical cadherins, such as E-cadherin [10]. These adaptor proteins, including: p120, β-catenin, and α-catenin, function to connect the F-actin filaments within neighboring cells to the cadherins in the extracellular space [10]. Desmosomes utilize non-classical cadherins (desmoglein and desmocollin) in conjunction with adaptor proteins (plakoglobin and plakophilin) to interact with intermediate filaments [11]. The primary function of the desmosome is to connect the intermediate filaments of neighboring cells to provide the extra strength and support to the epithelial layer [11]. 

There are two mechanisms for transport across the intestinal blood barrier: first is transcellular, and second is the paracellular pathway. In the transcellular pathway, solutes from the gut pass through the epithelial cells directly [10]. Transporters for different small molecules and metabolites produced in the gut are spread throughout the apical membrane of the epithelial layer [11]. This allows for the intake of the metabolites into the epithelial cells, which are then trafficked to the basal side of the epithelial cell and are deposited into circulation [11]. Paracellular transport of metabolites occurs in the extracellular space between epithelial cells [10]. This paracellular pathway is intracellularly regulated by cytosolic adaptor proteins as well as the formation of pores [10,11]. Specifically, for tight junctions, the claudins comprising the tight junctions are believed to form pores to selectively allow certain metabolites to pass [11]. Dysregulation of these different permeability pathways can lead to “leaky gut” diseases, such as celiac disease [10].

#### 1.2.2. Blood–Brain Barrier

For the blood–brain barrier, endothelial cells line the microvasculature and act as a highly selectively permeable barrier to prevent most metabolites and small molecules from passing through to the brain [12]. Due to the high selectivity of this barrier, the transcellular and paracellular permeability’s are highly regulated and generally controlled by two transporters, the efflux transporters and the nutrient transporters [12]. Similarly, the intestinal blood barrier there are cell adhesion junctions that connect the endothelial neighboring cells together, including the tight junctions, adherens junctions, and the desmosomes. Dysregulation of the transport of metabolites and small molecules across the blood–brain barrier can lead to a host of neurological diseases by losing selectivity or degradation of the barrier [12]. This would suggest a correlation to “leaky brain” disorders. 

Blood barriers are associated with most tissues and are a mucosal layer that prevents trafficking of large molecules, some small molecules, and bacteria across the permeable layer into the blood or other compartments. Two important permeable barriers involved with the microbiota–gut brain axis are the intestinal–blood barrier (IBB) and the blood–brain barrier (BBB). Disruptions of the normal function of these protective barriers has been attributed to a syndrome known as “leaky syndrome” [1]. Leaky syndrome has been associated with the gut in previous studies showing that dysbiosis leads to an increased permeability of the IBB, which allows bacteria and small molecule metabolites produced in the gut to travel to distal organs via the bloodstream [1]. In combination with leaky gut, leaky brain could lead to potential negative neurological consequences such as increased severity of depression, anxiety, or a host of neurological diseases [1]. Dysbiosis was shown to mediate increased permeability of the blood–brain barrier and the blood intestine barrier, which allows metabolites and hormones to more readily pass and leave the gut (IBB) as well as enter the brain (BBB) and alter the biochemical homeostasis of the brain [1]. The disruption of these barriers could be the result of cytokine stress, inflammation, or alterations to the gut flora [1].

The breakdown of the intestinal blood barrier has previously been shown to be a causative mechanism of “leaky gut” syndromes, such as celiac disease [1]. Similarly, recent studies have linked the breakdown of the blood–brain barrier to neuropathological conditions, such as Alzheimer’s disease [13]. As with “leaky gut” based syndromes, which are due to increased permeability of the intestinal–blood barrier, there is new evidence that this barrier breach can lead to brain pathology and the potential to become “leaky brain” conditions. A recent study implemented the use of a new cerebrospinal fluid biomarker to better understand the role of blood–brain barrier degradation in neuropathological diseases and disorders [13]. Through a combined use of this novel cerebrospinal fluid biomarker and magnetic resonance imaging, the investigators were able to elucidate the relationship between amyloid β and tau biomarkers (classical biomarkers of Alzheimer’s disease) and the breakdown of the blood–brain barrier [13]. In patients exhibiting early cognitive dysfunction there is a degradation of the blood–brain barrier combined with damage of the capillaries within the brain that occurs independently of the development of amyloid β or tau [13]. This indicates the potential importance of the detection of the blood–brain barrier integrity as a biomarker in the early diagnostics of neuropathologies, such as Alzheimer’s disease [13]. Patients could be diagnosed with a neuropathological conditions before developing clinical manifestations or the production of disease state proteins [13]. While most neuropathologies are do not form as a result of the blood–brain barrier degradation, “leaky brain” has the potential to exacerbate onset of neuropathologies.

### 1.3. MGB Axis: A New and Important Drug Target

As more knowledge of the relationship between the gut microbiota and the brain are uncovered, research focuses can begin to shift allowing these pathways to be used tohelp patients with neurological and psychiatric disorders. Antibiotic interventions are very disruptive for the gut flora and lead to dysbiosis, which can impact the metabolites produced in the gut [2,5]. This could ultimately have negative consequences for patients with an infection that also have depression, anxiety, or another mental disorder due to alterations of gut produced metabolites and hormones [14]. New methods are needed for treating patients while protecting their microbiota. 

One suggested method would be to engineer a genetic kill switch into the gut flora [2], this could be given to the patient via a probiotic and allowed to colonize the gut to aid in normal flora colonization. This approach could be useful in killing select members of the gut flora so as not to alter the chemical signaling between the brain and the gut, whilst still being able to clear the infection. An advantage of this approach would be that it would maintain the integrity of the host tissue [2]. Phage therapy is another strategythat could be effective for managing bacterial infections in the gut without altering the metabolites produced [5]. This approach utilizes a bacteriophage or bioagent produced by the phage to target and kill specific bacteria [5]. This technique utilizes a modified phage that is non-replicative and then the live phage is given orally to target the gastrointestinal infection [5]. Bacteriophages and bacteriosins in particular have a very narrow targeting range and therefore their lytic activity will not affect the normal gut flora or the host. Non-phage engineered approaches such as non-reproducing avidocins or bacteriosins, may offer superior utility and be able to safely eradicate infectious pathogens from food or hospitals without irradiation or harmful chemicals [15]. This method has advantages over phage technology due to its specificity for the bacteria and a non-lytic cycle. 

### 1.4. The Mycobiota and Microbiota across the Lifespan 

In embryonic, early development and prenatal and post-natal nutrition, the mycobiome and the microbiome, which encompasses many bacteria, are a key piece of the puzzle influencing metabolism, development, immunity, and behavior [16,17,18]. Not only do we still need to determine how much of the microbiome is fungi but we also need to better understand the types and their role among the other organisms. The mycobiome appear to be part of the rarer biosphere, which is poorly characterized [19]. It is still unclear how fungi influence the other members of the microbiome community and what influence they have. However, it is clear that fungi are becoming more of a threat as a cause of multi-drug resistant infections and pandemics. Therefore, it is important that we seek to understand their role not only within the environment but also within ourselves.

In addition, as we dynamically change the environment leading to microbiome evolution there is a renewed importance to understanding the mycobiome as we manipulate our own immune systems making the commensal species residing in the environment and our bodies a possible threat to our survival [20]. There is always a balance between the good and bad but any perturbation can tip the scales in favor of pathogenicity over symbiosis. Certainly, we live among fungal threats when we become immune compromised, have increased inflammation, or underlying defects that compromise our natural defenses. Fungi represent a challenge to detect and classify as well as to determine susceptibilities of and represent a growing threat in the wave of anti-microbial resistance [8]. 

Much work has been done on bacteria that effects early development but we are still working on understanding the fungal species that make up these diverse interactions. More work is needed to understand the origin of our fungal mycobiome and the way it changes and evolves as we age [20,21]. Fungal pathogenesis seems to be the most threatening at the beginning and end of life when we are most vulnerable. Likely, both overgrowth and perhaps undergrowth of these species cause related pathogenesis. The question still remains if there is benefit to establishing or re-establishing a mycobiome with pro-fungal agents. In addition, we ask what the impact of anti-microbial or anti-viral agents on this system could be. 

One approach is to consider routes of infection within the nervous system, such as those from blood-borne infections with *Candida albicans*, which is an increasingly recognized modern problem and may help explain increased proliferation and age-related activation of microglia and astrocytes in neurodegeneration and aging [21,22]. Further, it was recently shown by Nation et al., that a leaky blood–brain barrier and barrier breakdown is an early marker of cognitive dysfunction in humans [13]. Mouse models of low-grade candidemia have been used to monitor the effect of disseminated infection on cerebral function and relevant immune determinants. *C. albicans* yeast cells were found to directly cause a highly localized cerebritis marked by the accumulation of activated astroglia and microglia [23]. One protective mechanism seemed to involve Amyloid beta protein precursor which accumulates around granulomas, while cleaved amyloid beta (Aβ) peptide formation and deposition was found in this model. This group also showed CNS (central nervous system)-localized *C. albicans* was involved in transactivation of NF-κB and induced downstream production of interleukin-1β (IL-1β), IL-6, and tumor necrosis factor (TNF) [23]. Finally, mice infected with *C. albicans* display mild memory impairment that appeared to resolve with clearance of the fungal infection.

To find fungal elements we have to explore the natural reservoirs of where we find myco-organisms in the body such as gastrointestinal, respiratory, and reproductive systems. In addition, there is a constant push and pull between the pathogenic and non-pathogenic myco-organisms that exist both in our bodies and in the environment. These are affected not only by the general state of health but also factors like demographics, racial, ethnic, and gender differences [24]. Attempts to classify our fungal mycobiome, show a unique signature reflective of the individual, almost like a fingerprint. Differences in the host innate immunity as noted by Ghannoum et al., causesalterations in the oral fungal mycobiome between HIV positive and negative individuals as well asstudies involving the lung mycobiome between healthy and transplant patients [25]. 

There is evidence that the mycobiome may counteract some of the pathogenic species in our gut like *Clostridioides difficle*. Some studies suggest that the fungi compete with these pathogenic species and perhaps even mitigate their potential for harm [26]. Despite many studies about the vaginal microbiome, the fungal profile remains to be fully characterized [27,28]. There is much to learn about the mycobiome including its beginning in early life as well as its influence with inflammation and host immunity. 

Most of the research to date involves *Candida albicans* colonization. However, other endemic fungi like *Histoplasma* or *Aspergillus* are less likely to colonize and more likely to causedisease. Some organisms like *Cryptococcus* or *Pneumocystis* can co-exist within our bodies until we have a change in our immune system. *Saccharomyces cerevisiae* var. *boulardii* are considered the good fungi used in probiotics. There are different sites of the human body that serve as sites of colonization or reservoirs for fungi. *Candida*, in particular, has found many niches in the body. It has been shown to survive on both living surfaces as well as artificial devices via its ability to form biofilms and adapt to its host [29]. It is still unclear how it can transition from a component of the mycobiometo a pathogenic threat. This is undoubtedly affected by antibiotic resistance pressures, increased immune suppression in hosts, use of medical devices and prosthesis and other factors. These threats have issues in antifungal resistance in the form of species like *Candida auris*. No doubt there is a role for fungus in the inflammatory cascade, immune regulation, metabolism, and, in particular, the healthymycobiome. 

Likely, it may all start in our developing gut which is known to be affected by the type of delivery (vaginal versus cesarean), feeding (breast versus formula), antibiotics, environment, and gestational age. It is believed that the placenta mycobiome could be involved in immune responses impacting fetal immune system development, the course of maternal pregnancy and even future health outcomes for the newborn like diabetes and obesity [30]. Today, we seek to repair this dysbiosis by using pre- and pro-biotics together as synbiotics to repair the gut. We now know that no part of our development is truly sterile, even the womb, there are both inflammatory and infectious components that lead to outcomes like prematurity [31]. 

The role of the fetal intestine in both intra and extra uterine growth is still being explored. As it seems that pregnancy is really the initial exposure that the fetus has to the micro and thus, mycobiome, it may be possible to intervene in this development by dietary alterations in the mother such as the addition of certain additives to the diet [32]. Just how these relevant fungi are transmitted and how they remain colonized appears to be unclear (e.g., amniotic fluid exposure, method of delivery, and breast milk exposure) and more research needs to be done in this regard. The effects of these organisms can possibly explain the common complications of pregnancy such as pre-term labor or other inflammatory processes, as suggested by Neu et al. [33]. They may also extend their effects into the autoimmune life of the infant and adult, however, this has not been proven yet. Neu suggests that “the microbial metabolites and immunologic responses in mothers’ gastrointestinal tracts and genitourinary systems and mouth may have profound effects on the fetus.” Other relevant factors include method of feeding, exposure to antibiotics, as well as gastric acid exposure [33].

One particular area of interest includes the possible ongoing exposure to mother’s milk that breast-fed babies have as their sole dietary intake for at least 4-6 months of their life. It could be that this milk, which contains maternal bacteria and fungi, helps to build the micro and mycobiome of the growing baby [34]. As Thum et al. note, this dynamic environment may be altered by multiple factors, such as antibiotic exposure, allergens, and illnesses [32]. In addition, even less is known about the transfer of fungi to the infant. 

While it is frequently noted that breast milk is tailored to fit the needs of the infant, in a broader context, this could be interpreted as reflective of the maternal micro and mycobiome that is laid out specifically for her infant to help the infant survive and thrive in their external environment [35]. As Neu points out, the ramifications of the “milk microbiome” and in addition the milk mycobiome has yet to be fully explored [33]. In fact, Ward et al. did a study to characterize the skin, oral, and anal mycobiomes in 17 infants as well as the anal and vaginal mycobiomes of their moms by internal transcribed spacer 2 (ITS2) amplicon sequencing demonstrating that they differed by body site, mode of birthing and challenges the idea that transmission is exclusively vertical or sequential and not otherwise environmental (See Table A1 in the Appendix A) [36]. Ultimately, more studies are needed to evaluate the evolution and characteristics of our fungal mycobiome [37].

### 1.5. Autism and ASD in Relation to the Microbiota and Child Development 

Once weaned a child’s microbiota is stabilized towards an adult-type phylogenetic architecture and remains relatively stable throughout adulthood [3,6]. Due to the stable phylogenic persistence through adulthood our focus switches from the phylogenic architecture to dysbiosis [38]. Instead, we consider aberrant ecosystems and their associated diseases, as compared to unaffected individuals. The gut microbiota is increasingly implicated in the etiology of two common childhood developmental brain disorders, autism and autistic spectrum disorder (ASD). Although considerable conjecture centers on the pathobiology, one target is the microbiota–gut–brain axis [38,39,40]. The etiology of these broad spectrum disorders is unclear, but genetic and environmental factors play a role in their pathogenesis. It has been shown that maternal infection, the stresses of pregnancy, and a host of prenatal insults may increase the risk for neuro-developmental disorders later in life, such as schizophrenia, autism, or other cognitive and behavioral syndromes.

Several gastrointestinal abnormalities and high rates of antibiotic use are linked to microbial composition and contribute to the alteration of function found in ASD and autism-affected individuals [41,42,43]. In an ASD mouse model, studies under germ-free conditions show reproducible social deficits and increases in repetitive behaviors similar to those observed in ASD [44] and behavior traits that are autism-like. GI (gastrointestinal) phenotypes are also associated with altered microbiota and histone modification as well [40,45]. 

Many studies, support a role for the gut microbiota in the formation of the potential neuro-modulatory metabolite of phenylalanine and tyrosine, 3-(3-hydroxyphenyl) 3-hydroxypropionic acid (3-HPHPA) and possibly 4-(3-hydroxyphenyl) 3-hydroxypropionic acid (4-HPHPA), and others are involved in the pathogenesis of ASD, autism, dysbiosis, schizophrenia, and detected in human and rodent urine [46,47,48,49,50]. The psychobiotics, defined as probiotics that may improve psychiatric, or neurological illnesses supports a mechanism for autism and schizophrenia assertions. For example, a probiotic consisting of *Bacteroides fragilis* was given in early adolescence, which ameliorated some behavioral deficits in a rodent autism model [51,52,53]. Shaw et al, has made the assertion that several *Clostridia* bacterial species are at the heart of the schizophrenia and autism connection [48]. Our team recently reported this marker (HPHPA) in human CSF (cereberal spinofluid, serum, and in urine [50], confirmed by previous studies [54,55]. 

### 1.6. Consequences to Perturbation in the Gut Microbiota 

We argue that any factor that modulates the microbiota, such as antibiotic use, stress, disease or anything that prevents a child from receiving microbial aliquots from the mother, even during birth, can lead to permeability of the blood–brain barrier after birth and likely affect the transmission of virulence factors, pathogens or deleterious metabolites for an undetermined length of time. Besides metabolic disorders like obesity and gastrointestinal disorders like inflammatory bowel disease and irritable bowel syndrome, altered microbiota have also been linked to neuropsychological disorders such as depression and ASD [56]. Although it has been postulated, it is not well established that these factors indeed contribute to disease pathogenesis, in cases with autism, ASD or schizophrenia, and it remains to be determined. While the degree to which intestinal microbiota affects dietary preference is not well defined; the converse is evident in that diet can rapidly change the microbiome [57]. 

### 1.7. Aging and the Microbiota through the Lifespan 

In healthy individuals the normal gut flora is composed primarily of members of the Firmicutes, however, other phyla including Bacteroidetes, Proteobacteria, and Actinobacteria, which combined with the Firmicutes comprise approximately 90 percent of the gut flora [58]. Microbial phylogenic architecture with aging has large inter-individual variability [4,7] and reduced biodiversity in older individuals, which compromises microbiota stability compared to younger individuals [4]. Over the course of the human lifespan, shifts in the diversity and abundance of the different phlya occur in response to factors such as diet, infections and other related factors (See Table A2 in Appendix A). While dietary habits, can play an important role the question is whether inflammation plays a more significant role. Aging is characterized by immunosenescence and thymus involution, which results in the loss of T-cell education and functional deterioration of the neuro-immune system [59]. Aging affects the human gut microbiota in several ways, namely, changes in the composition of metabolic byproducts derived from host microbiota and mycobiota, changes in the phylogenetic composition, and interaction with the immune system [60]. These changes are associated with immunosenescence and inflammatory aging and the age-related diseases associated with these processes. Bacterial and fungal infections like blood-borne infections with yeast such as *C. albicans* are an increasing problem, with age and may be concomitant with immunosenescence [61,62]. 

We speculate that yeasts could be involved in chronic neurodegenerative disorders, for example, plaques in mice resemble senile plaques of Alzheimer’s disease (AD) patients. Previous studies found that yeasts are commonly found in the brains of AD patients. Yeast deploy a mechanism to avoid the usual defenses, especially the blood–brain barrier, that produce immunologic defense mechanisms causing cytokine stress, down-regulation of inflammation associated genes, and improvement in colonic mucosal conditions. Novel new probiotics could be used in the promotion of longevity and prevention of immunosenescence. They offer new targets and treatment strategies for improving the health of older individuals who already may be immunocompromised. In this context, the delivery of probiotics and prebiotics or synbiotics may be useful for both prevention and treatment of age-related pathobiologic conditions from the improvement of immune function to the maintenance of the integrity of the gut barrier function, and the down-stream blood–brain barrier function [63] prevent most infectious organisms from entering the brain, particularly the parenchyma [1]. 

### 1.8. The Promise of Fecal Transfer from Young Donors to Old-Aged Recipients

Microbial populations reveal significant change in several spheres of aging, and that gut microbes have a significant influence on the health of the elderly [60]. In that regard, a novel target first identified by aquatic researchers involves delivering gut microbes from young killifish to older killifish thus extending the life span of the older fish, which is analogous to vertebrate parabiosis where specific changes that occur between two animals sharing a circulatory system will reverse some aging parameters in the older animal. How this finding in the short-lived African turquoise killifish (*Nothobranchius furzeri*) extrapolates to mechanisms of higher vertebrates is still unclear [64]. Analogously, we have established protocols for fecal transplant though anaerobic encapsulation of lyophilized material [65]. It may be as simple as repopulating appropriate commensal species or a “eumicrobiota” to coin a concept, but a better understanding could help the elderly by imparting beneficial metabolites and a helpful second genomic contribution within us all. 

This work is reminiscent of older parabiosis experiments in rodents where young and older animals are conjoined through a common circulatory system [66,67,68]. The autors of the work [60] show interventions in this distantly related vertebrate model using microbiota from young did prevent decreases in species diversity associated with aging and maintained a phenotypically younger gut bacterial community. These authors characterized the young phenotype as overrepresentation of Exiguobacterium, Planococcus, Propionigenium, and Psychrobacter [60]. 

The effect of the aforementioned process, which we have used in human transplant [65] was long-lasting and systemic with beneficial effects including extended life span in this vertebrate model. One often-overlooked research area, concerning the elderly and microbiota, is the significance of microbial co-metabolism on drug efficacy or toxicity involving bacterial enzymatic activity and drug metabolism [2]. This is especially important for personalized medicine and the need for a new physiology of the elderly, who often use multiple drugs for extended periods of time compared to the young. The microbiota-drug interaction will likely involve manipulating the mammalian gut microbiome to enhance health, such as producing a pro-drug or vitamin that can reverse age-related damage to DNA or aid cellular repair mechanisms from DNA damage, such as nicotinamide mononucleotide (NMN). 

Mice administered NMN were found to live 20 percent longer and were able to run faster than the age matched control cohort [69]. Moreover, cells were indistinguishable from the young mice in all respects except for repair findings. There are already a great many ways to influence the gut microbiome, including calorie restriction, and the observed impact of these strategies puts some limits on what it is plausible to expect from a more rigorous, informed, and technologically assisted adjustment of the microbial population. Species with short lifespans also have a much greater plasticity of longevity compared to humans in interventions of this manner. The method of calorie restriction that increased the life span of mice by 40 percent are not nearly as effective or beneficial in humans. 

### 1.9. Methionine Cycle Genes Contribute to Epigenetic Regulation

Bacteria could provide missing enzymes and correct genetic mutations in metabolism, such as the common methylenetetrahydrofolate reductase gene (MTHFR) Genetic polymorphisms in the MTHFR gene results in low methionine production and high homocysteine (HCys) levels, which in turn may lead to accumulation of S-adenosylhomocysteine (SAH) and low levels of S-adenosylmethionine (SAM) in humans where homocysteine contributes to cardiovascular disease and hyperhomocysteinemia induces oxidative stress. MTHFR is important to the methionine cycle and transsulfuration pathway that regulate SAM, an important methyl donor for histone methyltransferases (HMTs) and DNA methyltransferases (DNMTs) [70]. Mutations in the MTHFR gene produce low levels of SAM and the MTHFR enzyme reduces 5,10-methylenetetrahydrofolate to 5-methyltetrahydrofolate, necessary for the transference of a methyl group to the methionine [70]. Low levels of SAM can affect global DNA and histone hypomethylation in part through hyperhomocysteinemia, making this treatment an important possible means to affect epigenetics via modulation of nutrient or vitamin cofactor uptake.

Analogous to NMN for DNA repair, H_2_S, a significant bacteria-derived gaseous water and lipid soluble signaling mediator, could be used to protect cells from oxidative stress and ischemia-reperfusion and other insult. Similar to nitric oxide or carbon monoxide, H_2_S plays a role in both physiologic and pathophysiologic mechanisms from regulation of synaptic transmission, to inflammation, vascular tone maintenance and angiogenesis for vascular conditions. Further, the hydrogen sulfide pathway is involved in cancer and could be a suitable treatment target or be developed as a drug or drug delivery vehicle through novel probiotics. Using bacterial biosynthetic machinery, we could envision dietary l-methionine, l-cysteine, and the cofactor pyridoxal-5′-phosphate to produce H_2_S in tissues or the gut for therapeutic purposes. Deficiencies would be the first issue to correct through dietary means, such as those involving the transsulfuration pathway and increasing the synthesis of glutathione (GSH) [71]. Arguably, GSH is the most significant mammalian non-enzymatic antioxidant, especially important for de-poisoning the host from mercury contamination, without use of questionable chelation therapy approaches.

Outside of affecting nutritional deficiencies or medical conditions that are due in part to gut pathogen dysbiosis, it seems unlikely that large gains in the human lifespan are going to be realized. However, the deleterious aspects of aging may be slowed or even partly reversed with these approaches if enough translational funding opportunities become available. We could certainly expect an improved quality of life and maybe a longer health-span from these technologies. In this regard, a few concepts should be mentioned starting with Caleb “Tuck” Finch’s idea of negligible senescence, which was further refined by Aubrey de Grey as Strategies for Engineered Negligible Senescence (SENS), aimed at developing a better understanding or even “cure” for aging [72]. Perhaps the rate of achieving these strategies would be hastened though the microbiota and its tremendous potential for revolutionary transformation. This new approach will likely culminate in novel therapies and technologies capable of repairing the known and future forms of accumulated cellular and molecular age-related damage. 

Finally, the apparent low-hanging fruit on the vine of science would involve using typed and cross-matched synbiotics as a novel “new drug”. This concept is best illustrated by a recent case where a woman of relatively normal weight with *Clostridioides difficile* infection, was successfully treated with a fecal transplant with widely different gut bacteria while concurrenty experiencing rapid weight gain of about 50 pounds [73]. The authors did not offer a hypothesis for the mechanisms of transmission nor identify and sequence the species involved over the course of the treatment of the obese phenotype, thus not demonstrating a clear causal relationship in this case. Nevertheless, it is inferred that the microbiota plays an important role in weight gain [74,75] and could help in treating obesity, diabetes, heart disease, and a host of other common illnesses. 

### 1.10. CRISPR and Gene Engineering Prospects

A newer approach for the transformative potential of the microbiota comes from clustered regularly interspaced short palindromic repeats (CRISPR), which was first identified in bacteria and function as a prokaryotic immune mechanism to bacteriophage infection. Instances of viral infection are met with Cas cleavage of phage DNA which is reintroduced to the bacterial genome at the CRISPR loci as spacer regions flanked by palindromic repeats. This natural bacterial defense mechanism has been engineered to become a powerful genome editing tool using synthetic guide RNA to allow sequence specific nuclease activity. The use of short RNA guides differentiates it from other synthetic biological techniques, namely transcription activator-like effector nucleases (TALENs) and zinc finger nucleases (ZFN). The latter rely on specifically engineered DNA binding domains that are limited to the single set of nucleic acid sequences they are designed for, but are prone to off-target mutagenesis [76]. 

In contrast, the more easily synthesized and interchanged CRISPR RNA (crRNAs) guides afford the CRISPR-Cas system greater versatility and flexibility, especially when multiple modifications of many different genomes is desired. The use of multiple recognition sequences on single crRNAs allow simultaneous editing of multiple sequences of interest [77]. Additionally, nuclease-null Cas9 can similarly bind crRNA and recognize target sequences without excising DNA, allowing it to regulate transcriptional activity, function as nucleic acid tethers, labeling molecules, and DNA scaffolding. This affords CRISPR genetic manipulation powerful control over bacterial genes and metabolism through target translational activation and inhibition [78]. 

The development of CRISPR-Cas as a genetic engineering tool has been paralleled by growing understanding of the human microbiome and its far-reaching consequences for human physical and mental health. Modern dietary and environmental factors can have disrupting influences on the microbiome and by proxy, health outcomes. Medications and dietary habits have been shown to cause changes in populations of microbiota [79]. 

Studies of humanized mice demonstrate the detrimental effect these diets have on gut biodiversity. While fecal transplants are effective in restoring normal populations of microbiota, continued consumption of low fiber and high carbohydrate foods will cause repeated dysbiosis in many gut microbes. Barring population wide dietary changes towards high fiber and complex starch foods, it may simply be more feasible to engineer bacteria prone to decline under these conditions using CRISPR to enable microbiota to better adapt to our modern dietary and pharmacological environments.

CRISPR inhibition (CRISPRi) and activation (CRISPRa) have been used to modulate bacterial genetic expression. Here, designing is particularly useful for genes encoding bacterial pathogenic factors, such as biofilm formation and toxin eradication. CRISPRi has been used to inhibit transcription of mRFP in *E. ecoli* through horizontal gene transfer of plasmid containing dCas9 and appropriate short guide RNA (sgRNA) [80]. Similar techniques were employed to synthesize sgRNA targeting the *E. coli* luxS gene which encodes for a synthase involved in AI-2 which guides biofilm formation [81]. In principle, given appropriate sgRNA and delivery, any gene of interest can be subject to CRISPR activation or inactivation.

Neurodegenerative diseases such as Parkinson disease (PD) and Alzheimer’s disease are associated with dysbiosis of microbiota in a bidirectional relationship typical of the gut–brain axis [82]. Microbiota are critical in producing signal molecules and hormones from the intestinal metabolism which can affect neurological disease states and central nervous system inflammation. Additionally, disruption of the microbiome can further propagate neurodegenerative disease pathobiology [83,84]. With PD and AD signaling effector molecules, in the microbiome are an important for understanding mechanisms underlying disease pathogenesis and using the microbiota is one targeting approach for potential treatments for these disorders. 

As discussed, CRISPR allows for influence over bacterial metabolism through either modification of whole genes, or nuclease lacking CRISPRi/CRISPRa binding to target genes in order to effect gene expression without sequence modification. Genetic manipulation, especially with respect to bacterial metabolism, signal molecule production, and hormonal secretion can offer a much more direct approach to influencing CNS inflammation, especially as a result of dysbiosis found in neurodegenerative disorders. 

Prebiotics, such as, short chain fatty acids (SCFAs), are a bacterial metabolite of particular importance as they act as both signaling molecules and provide available energy from otherwise indigestible carbohydrates, and play a role in maintaining intestinal cell layer integrity [1,85]. SCFAs act as signal molecules and have regulatory effects on G-protein-coupled receptors (GPCRs) and histone deacetylation (HDAC). GPCRs, their cognate kinases and HDACs are implicated in the pathogenesis of neurodegenerative diseases [86,87]. Among other signaling effects, SFCAs modulate immune responses by activating GCPRs, which is a very common drug target, suppressing inflammatory responses [88]. HDACs also play a role in neurological and inflammatory pathogenesis, particularly in alpha-synuclein toxicity [89]. Because of their role on Lewy body formation, alpha-synuclein suppression has been an important approach for PD and AD therapeutics [90]. SFCAs inhibit HDAC activity and can thereby modulate alpha-synuclein pathogenicity [91]. 

Because of their importance as signaling and receptor molecules, enhancing SCFA availability appears important in addressing neurodegenerative disease [92]. In that regard, the prebiotics that include the short-chain fatty acids and bacterial products of fermentation are implicated in ASD and its treatment [93,94]. By enhancing SCFA production through the metabolism of intestinal carbohydrates and dietary fiber, there is the potential to affect HDAC inhibition, lower alpha-synuclein toxicity and inhibit Lewy body accumulation. To date, there are few treatments and inadequate diagnostic tools for diffuse Lewy body disease. Additionally, activation of GPCRs through increased SCFA availability can contribute to reduced inflammatory responses. CRISPR modification of carbohydrate metabolic pathways through simple changes such as ribosomal binding site (RBS) optimization can increase rates of carbohydrate fermentation and SCFA production and have an inhibitory effect on CNS inflammation and Lewy body formation. Additionally, CRISPRa activation of target genes can similarly increase transcription and SCFA availability. Finally, a new synbiotic combination using probiotics with dietary SCFA’s, fiber, high protein, and vitamin D could be a bariatric support regimen for gastric bypass and the like.

### 1.11. Novel New Antibiotics and Synbiotics to Treat Diseases and Age-Related Syndromes

Probiotics and synbiotics have been proposed as potential solutions to dysbiosis and could restore microbial balance, eventually positively affecting metabolism and signal molecule production with respect to the gut–brain neuroendocrine axis [2,82]. Though live culture dosing offers a simple and efficacious approach [95], new anaerobic bacterial probiotics are needed. When coupled with modern molecular and synthetic biology techniques, particularly CRISPR, these probiotics could offers far more powerful tools to address the bidirectional effects of neurodegenerative disease co-metabolism. 

A hurdle to those of us who have considered and devised these approaches is the requirement to label this treatment as a drug. The microbes used are within the public domain and are commonly found in the gut of humans throughout the world. Understandably, there is a need for safety and oversight considerations with microbes, but stringent regulations can restrict and slow the development of these treatments, relegating them to interests that have the most money. One solution may be to create a new oversight and regulatory agency for the rapid and safe development of these transformative technologies for the benefit of human kind.

### 1.12. Ruterin as a Natural Source of Antibacterial Agent.

Ruterin is a complex mixture of glycerol-derived products, consisting of acrolein, acrolein hydrate, 3-hydroxypropanal (3-HPA), and its dimerization product, 3-hydroxy-2-(3-hydroxypropyl)-1,3-dioxolane among other not yet identified components [96,97]. Ruterin is a metabolic product of glycerol and is excreted as an antibacterial agent by many symbiotic gut-bacterial species and it exerts antibacterial effects against a broad variety of Gram-positive and Gram-negative bacteria, bacterial spores, and also has potential antifungal effectiveness [98]. The dehydration of glycerol to generate 3-HPA is effected by a coenzyme B12-dependent glycerol dehydratase, as established in cases of some *Lactobacillus* species [99,100,101,102,103,104]. The involvement of free-radical mechanism for this dehydratase catalyzed formation of 3-HPA was shown by ESR techniques [101,102,105]. 

Although, the 3-HPA is rapidly further reduced to 1,3-propanediol by 1,3-propanediol dehydrogenase, in certain microbial species, including *Lactobacillus ruteri* and in *Eubacterium hallii*, there is a lack in dehydrogenase activity leading to the secretion of 3-HPA by these organisms. The term ruterin is derived from its major source of production, *Lactobacillus ruteri*. Ruterin exhibits antimicrobial properties and is involved in many beneficial effects, such as cancer suppression and attenuated inflammation. The antimicrobial properties of ruterin against *Escherichia coli* strains was demonstrated by co-administration of *Lactobacillus ruteri* and glycerol to the bovine gastrointestinal tract, a major reservoir of the *Escherichia coli*, and a possible cause of food contamination [106]. The biofilm of a Gram-positive bacterium, *Lactobacillus ruteri*, on dextranomer microspheres was shown to be effective for targeted delivery of ruterin [107]. 

The ruterin complex is formed through both enzymatic and nonenzymatic reactions, and its major constituents, such as 3-HPA, are in dynamic equilibrium with other components. Abundant evidence on the biosynthetic pathways for the formation of ruterin indicates that in initial stages, glycerol dehydratase transforms glycerol into the 3-hydroxypropananl (3-HPA), which is in equilibrium with acrolein that is formed through further nonenzymatic dehydration path (Figure 1). 3-HPA is also in dynamic equilibrium with its dimeric 1,3-dioxolane product (3-HPA dimer) and with its hydrate. The abundant quantities of 3-HPA available from the biomass would also provide a source of value-added chemicals, possibly as alternatives to the fossil feedstocks [108].

The involvement of acrolein as the constituent of the ruterin system, and its direct involvement in the decontamination of the ruterin-sensitive gut bacteria was established through a combined experimental technique involving ion-exclusion chromatography with pulsed-amperometric detection (IC-PAD), NMR and ultra-performance liquid chromatography–electrospray ionization mass spectrometry (LC/ESI-MS) [96]. It is also likely that acrolein may exist in dynamic equilibrium with a heterodimeric 1,3-dioxolane arising from its reaction with 3-HPA. Perhaps through the latter stabilizing effect, the reactive acrolein is generated as it is needed for its reactivity with thiol antioxidants, such as glutathione. Thus, by depleting the intracellular sources of antioxidants that would sequester the reactive active-oxygen and reactive nitrogen species (ROS and RNS) ruterin contributes to exacerbated oxidative stress and the resulting microbial toxicity. 

Ruterin is also involved in the detoxification of the otherwise toxic heterocyclic amines, such as imidazopyridine compounds (PhIP), which are formed as a result of the Maillard reactions of reducing carbohydrates and amino acids or proteins. The observation of the acrolein conjugation of PhIP provides strong evidence of the potential antibacterial effects of the acrolein component of ruterin [96,109]. Extracts of ruterin were also shown to be effective in the reducing plate counts of *Enterobacteriaceae* and yeasts in minimally processed fresh lettuce [110].

## 2. Conclusions and Outlook

While 3-HPA not only serves as a source of the endogenously formed antibacterial agent acrolein as shown by Lacroix et al. [96], but it is likely to stabilize acrolein through a possible acetal product that would serve as a slow-release source of the acrolein for its antibacterial effects (Figure 1). Thus, acrolein-3-HPA dimeric product may be an important component of the antibacterial ruterin. The latter potential stabilizing effect of 3-HPA on the acrolein (which in relatively large concentrations is a known carcinogen) warrants further examination. Thus, the toxicity of the ingested acrolein-rich foods, those arising from processed foods, may be ameliorated by the gut-bacteria derived ruterin system. Furthermore, we hypothesize that various 1,2-dicarbonyl precursors of advanced glycation end products (AGEs) may also be in dynamic equilibrium with the 3-HPA so that 3-HPA exerts its anti-glycating effect by sequestering the AGE-precursors. 

We have used directed evolution to mutate *Pseudomonas* sp. after discovering this species could metabolize diesel fuel spills [111]. This hypothesis, although not yet tested in the gut microbiota, has proof of principle behind it and will support a possible pathway for ameliorating the toxic effects of the exogenously consumed and endogenously formed AGEs. Similarly, one hope is that bacteria one day can be designed to target and reverse the age-related crosslinks in tissues protein and nucleic acids or lipids. The earliest expectation is to succeed by deglycating non-enzymatic post translational modifications from sugars by targeting the early reversible glycation intermediates and preventing downstream propagation and advanced end product formation [112,113,114,115,116]. At the same time, we explored directed evolution and phage display technology and other engineering mechanisms, such as inhibitors of bacterial enzymes. It is possible for phage display coupled with a kill switch to target many bacteria and toxins or novel substrates, such as now being considered for the treatment for cancer. Here, the phage could reach its target and carry with it a wavelength-specific light-activated killing mechanism which can be both target and a treatment. In summary, when one considers the prospects of the microbiota and mycobiota as a transformative force for change against the hype that may come with expectation both perspectives have a hydraulic effect on the international scientific system and community. As eternal optimist’s we hope to have objectively argued for transformative properties of the microbiota and mycobiota.

## Figures and Tables

**Figure 1 biomedicines-07-00024-f001:**
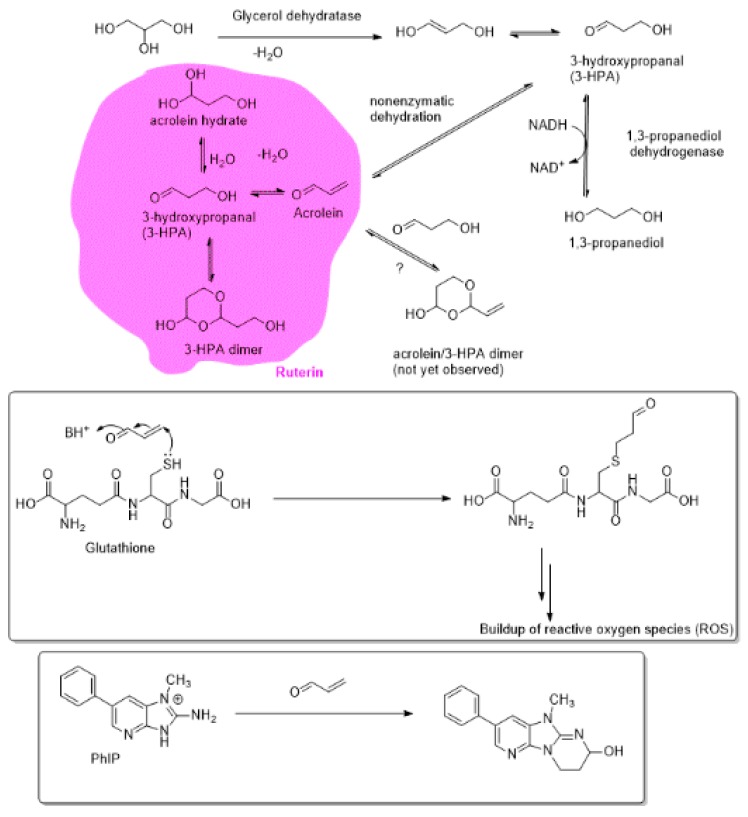
Formation of the Ruterin, a composite mixture of acrolein, acrolein hydrate, and 3-hydroxypropanal (3-HPA) dimer; and the Michael reaction of acrolein with glutathione and imidazopyridinium compounds (PhIP), which are constituents of the ruterin-susceptible bacteria.

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
