# Peer review of "The Transformative Possibilities of the Microbiota and Mycobiota for Health, Disease, Aging, and Technological Innovation"

_biomedicines, 2019, doi:10.3390/biomedicines7020024_

Reviewer 1 Report

This concept paper discusses the intriguing topic on microbiota and mycobiota and summarizes a range of published work related to the transformative possibilities of the microbiota and mycobiota for health and diseases. Moreover, the authors have introduced the concept of mycobiota and microbiota-gut-brain neuroendocrine axis and its role in inflammation, immune regulation, and metabolism. It has been written in a clear and comprehensive way and reflects the purpose of the manuscript. However, some minor edits may improve the quality of the CONCEPT PAPER.

Authors have cited review “Leaky Gut, Leaky Brain?” multiple times in this review manuscript. Authors should consider citing original citation throughout the review manuscript, not the review manuscript. Few Examples

Line #132 Author claiming that “… our previous work showing that dysbiosis…….[1].

Line 134: Line # In combination with leaky gut, leaky brain could lead to potential negative neurological consequences such as increased severity of depression, anxiety, or a host of neurological diseases [1].

Line # 141: The breakdown of the intestinal blood barrier has previously been shown to be a causative mechanism of ‘leaky gut’ syndromes, such as celiac disease [1].

Line#355 : …….prevent most infectious organisms from entering the brain, particularly the parenchyma [1].

Author should cite the original and not “Leaky Gut, Leaky Brain” review. This is true about more citation in this review.

Line #86: Author is citing [102] reference under heading “Blood barriers and their roles in the MGB axis” which is suddenly after reference # 1-9. Can author explain the same? 

Line # 165 Author should consider adding reference …….metabolite and hormones (reference…)

Line # 177-179 : Author should consider adding reference  …..harmful chemicals (reference…)

Line # 185 ………………… immunity and behavior ( reference)

Line# 236 Despite many studies ……….??

Line # 344: need reference

Line # This has echoes of parabiosis experiments in mice, linking younger and older animals together through…. have been copied from https://www.fightaging.org/archives/2017/04/gut-microbes-from-younger-killifish-extend-life-in-older-killifish/

Line #395-399: Author should add reference

Line #425-428: Author should add reference

Line #554 : Escherichia coli should be Italics

Line # 573: the reactive active- ???

Line # 597: ……………..diesel fuel spills  (Ref…??).

Line #603: as evidenced by ……..seems incomplete.

The authors may consider adding a new table for summarizing the study on the role of fungus (mycobiota) on various human diseases. Although information is provided in the content of the review, adding a new table will heplful for the readers.

It needs to be proofread for proper English grammar and small mistakes

Author Response

We wish to thank the reviews for their helpful comments. We are following several suggestions and making a table for the microbes etc. We are waiting on a student, who could not address issues due to midterms. The final version will be submitted no later than CB on Tuesday at the latest.

We have reread each section and one author has unified the english and syntax.

As for the Leaky Gut, Leaky Brain,I understand we cite data ares, but this is my new concept and it has never been stated in the literature, therefore, my review is the only and most concise source available. There are some supportive  findings that lead me to the conclusion and synthesis.

The reference program had some issue. We are correcting this with a different program.

Reviewer 2 Report

The present manuscript as a Concept Paper was written well. Reference numbers were not in order of appearance. Correct them.

Author Response

We wish to thank both reviewers for their helpful critiques and now conclude our rewrite. It is attached below and addressed in tracked format what we did to improve the submission including making 2 tables as suggested and rewriting the sections as suggested that I had commented on in online print. as well as adding citations and new references.

Dr. Obrenovich
